# Validating the Definition of Lumbar Instability—A Cross-Sectional Study with 420 Healthy Volunteers

**DOI:** 10.3390/jcm13206116

**Published:** 2024-10-14

**Authors:** Manabu Suzuki, Yasuhisa Tanaka, Ko Hashimoto, Takumi Tsubakino, Takeshi Hoshikawa, Kohei Takahashi, Myo Min Latt, Toshimi Aizawa

**Affiliations:** 1Department of Orthopaedic Surgery, Tohoku Central Hospital, Yamagata 990-8510, Japan; manabu1029@hotmail.com (M.S.); ytanaka@tohoku-ctr-hsp.com (Y.T.); ttubakino@tohoku-ctr-hsp.com (T.T.); 2Department of Orthopaedic Surgery, National Hospital Organization Sendai Medical Center, Sendai 983-8520, Japan; 3Department of Orthopaedic Surgery, Tohoku University Graduate School of Medicine, Sendai 980-8574, Japan; yojuhei@gmail.com (K.T.); toshi-7@med.tohoku.ac.jp (T.A.); 4Sendai Orthopaedic Hospital, Sendai 984-0038, Japan; thoshikawa@hotmail.com; 5Mandalay Orthopaedic Hospital, Mandalay 11101, Myanmar; myominlatt.dr@gmail.com

**Keywords:** lower back pain, lumbar instability, dynamic radiography of the lumbar spine, intervertebral range of motion, sagittal translation distance, criteria

## Abstract

**Background/Objectives**: Low back pain is thought to be caused by lumbar instability. To date, multiple definitions of radiological lumbar instability have been used without verifying the “normal range” of the lumbar segmental mobility. Ideally, normative data for lumbar mobility in healthy individuals are required to establish an acceptable threshold for lumbar instability. This study aims to elucidate (i) the prevalence of so-called radiological lumbar instability at each lumbar spine level in conventional criteria and (ii) a practical radiological threshold for lumbar instability in healthy individuals. **Methods**: Participants completed a questionnaire and underwent standard active dynamic radiography of the lumbar spine in the standing position. Intervertebral range of motion (IROM) and sagittal translation distance (ΔST) were measured at each intervertebral level. Nachemson’s criteria of radiological lumbar instability were applied. **Results**: This study involved four hundred and twenty participants (249 males and 171 females); 76% (320/420) met the criteria for radiological lumbar instability. The definition of lumbar instability based on IROM and ΔST was achieved by 0.2% and 1.7% of participants at the L5–sacrum (L5–S) level, respectively. **Conclusions**: The normative data of lumbar mobility were obtained from a large number of participants who had less LBP-related ADL disability. The widely accepted criteria for lumbar instability were not applicable except for the L5–S level. Further studies of lumbar mobility, including patients with severe LBP, might prove the relationship between hypermobility of the lumbar spine and LBP.

## 1. Introduction

The lifetime prevalence of low back pain (LBP) is reported to be 58–84% [1,2,3,4]. As such, LBP is a major socioeconomic burden worldwide. The annual economic loss due to LBP is estimated to be JPY 836.5 billion (approximately USD 5.9 billion) in Japan [5]. The main causes of LBP are trauma, neoplasms, infection, lumbar spondylolisthesis, lumbar spinal canal stenosis, lumbar disc herniation, and spine deformity [6], as well as lumbar instability [7,8,9]. Out of them, lumbar instability has been believed to be one of the major causes of LBP, justifying fusion surgeries for the pathology [10,11,12]. There have been various studies defining radiological lumbar instability by investigating the precise alignment of the lumbar spine in a limited number of healthy individuals to date [13,14,15,16,17]. Among various definitions for radiological lumbar instability, Nachemson’s criteria [18] have been widely used in many studies for decades. Surprisingly, these criteria proposed several decades ago have not been updated to date, and no studies have yet validated the feasibility of these definitions in large populations. In this point of view, normative data for lumbar mobility in healthy individuals are required to establish an acceptable threshold for lumbar instability. Lumbar instability is identified from lumbar dynamic radiographs by measuring various radiological parameters at each intervertebral level [18,19], most frequently, the intervertebral range of motion (IROM) and sagittal translation distance (ΔST) of vertebral bodies measured in lumbar lateral flexion–extension radiographs. This study aimed to determine (1) the prevalence of so-called radiological lumbar instability at each lumbar spine level according to conventional criteria and (2) establish a practical radiological threshold for lumbar instability in healthy individuals.

## 2. Materials and Methods

### 2.1. Study Design

A cross-sectional observational study with a diagnostic accuracy design was conducted following the STROBE guidelines [20]. The study participants included healthy volunteers who underwent a periodical medical examination at our hospital between January 2018 and June 2018. This study enrolled participants who agreed to undergo an optional lumbar dynamic radiological examination after signing written informed consent. Because of the characteristics of the hospital’s governing board, 75% of participants were schoolteachers. The rest were public employees, office workers, unemployed persons, housewives, self-employed persons, and farmers. This study excluded participants with a spinal surgery history, spinal trauma and infection, scoliosis exceeding 20°, isthmic spondylolisthesis, transitional vertebrae, poor quality radiographs for measurement, and inadequate/incomplete answers from the questionnaire. A completed questionnaire and standard dynamic lumbar spine radiographs were collected from each participant. Participants were requested to actively flex and extend their bodies as much as possible in the standing position for the dynamic radiographs. A standard tube-to-film distance of 100 cm was used. The radiation exposure for dynamic radiographs was approximately 0.4 mSv per participant.

The questionnaire included basic information, medical history, Japanese versions of the Oswestry Disability Index (ODI), and Roland–Morris Disability Questionnaire (RDQ). ODI ranged from 0% to 100%, and RDQ score from 0 to 24 points; for both, a higher score indicated more disability in an individual’s daily life [21]. The analysis excluded participants with an ODI score of ≥20% and RDQ score of ≥14 points to obtain data from nearly normal individuals with fewer disabilities in LBP-related activities of daily living (ADL) [22,23].

This study used Nachemson’s definition of radiological lumbar instability [18]. IROM and ΔST were measured and calculated for each intervertebral level between L1–2 and L5–sacrum (L5–S), as shown in Figure 1. The measurement of ΔST was similar to a previous study [24]. A histogram by intervertebral level displays the IROM and ΔST distribution of the participants. Based on a previous report [18], radiological lumbar instability was confirmed when at least one of the intervertebral levels between L1–2 and L5–S met any of the following criteria: (i) IROM of ≥10° from L1–2 to L4–5 or ≥20° at L5–S; or (ii) ΔST of ≥3 mm at L1–2 to L4–5 or ≥4 mm at L5–S.

The number of intervertebral levels meeting the criteria of radiological lumbar instability based on IROM and ΔST for quantitative evaluation of lumbar instability was defined as IROM and ΔST scores, ranging from 0 to 5. The proportion of participants was plotted in bar graphs by IROM and ΔST scores.

Finally, two board-certified spine specialists with >15 years of clinical experience (M.S and K.H) conducted the imaging analysis using a workstation software (EV Insite^®^ version 3.14.1.10, PSP Corporation, Tokyo, Japan). M.S. measured radiological parameters (intervertebral disk angle and sagittal translation) on dynamic radiographs of 100 participants twice with a 6-month interval between measurements to assess the reliability of measurements. Furthermore, K.H. independently measured the same radiographs in the same way. Then, the intra- and inter-rater reliability of intervertebral disk angle and sagittal translation were calculated.

### 2.2. Statistical Analysis

The sample size was set to estimate the upper limit of a 95% confidence interval with appropriate accuracy. More than 300 cases were needed to estimate the 95% confidence interval with an accuracy of ρ = 0.1 for the upper limit of 95% confidence interval. The refusal rate for participation in our study was expected to be 50%, so the recruitment period was reserved for from 600 to 700 participants visiting our hospital.

Data were collected in Microsoft Excel sheets (Microsoft Excel version 2409, Microsoft Corporation, Tokyo, Japan) and exported to Statistical Package for the Social Sciences Statistics version 28 (IBM Japan, Tokyo, Japan) for statistical analysis. The proportion of participants meeting the criteria for lumbar instability was calculated for each radiological parameter and intervertebral level. The standard upper limit of IROM and ΔST was defined and calculated as the mean plus two times the standard deviation at each intervertebral level. Intra- and inter-rater reliability were evaluated with intra-class correlation coefficients. ICC (1, 1) and ICC (2, 1) were used for intra- and inter-rater reliability, respectively. Landis [25] criteria were used to interpret ICC agreement values: slight (r = 0.00–0.19); fair (r = 0.20–0.39); moderate (r = 0.40–0.59); substantial (r = 0.60–0.79); and almost perfect (r = 0.80–1.0) reliability.

## 3. Results

Of the 697 individuals who visited our hospital during the enrolment period, 119 declined study participation. Finally, 420 participants (249 males and 171 females) were included after excluding 158 participants. The median age of the participants was 55 years (range: 24–83 years). Table 1 shows the demographic data. Figure 2 shows the histograms of ODI and RDQ scores, respectively. Of the participants, 26% had an ODI of 0%, and 77% had an RDQ score of 0.

Figure 3 indicates the distribution of participants’ radiological parameters at each intervertebral level. ΔST at the L5–S level was <1 mm in 65% of participants. Conversely, 76% of participants (320/420) met the criteria for radiological lumbar instability. Among them, 4% (18/420) demonstrated instability with IROM alone, 51% (216/420) had instability with ΔST alone, and 21% (86/420) demonstrated both.

Figure 4 indicates the proportion of participants with lumbar instability at each intervertebral level. At each level, except for L5–S, >20% of participants have radiological lumbar instability based on ΔST. Only 0.2% and 1.7% of participants met the criteria for lumbar instability at L5–S based on IROM and ΔST, respectively. The mean IROM value in healthy participants ranged from 4.3° to 5.9°, and the standard upper limit ranged from 10.3° to 13.9° at each intervertebral level. The mean ΔST value in healthy participants was 0.9–2.7 mm, whereas the standard upper limit of ΔST was 3.3–5.5 mm. Table 2 shows the mean, standard deviation, and standard upper limit of IROM and ΔST. ΔST was smallest at L5–S among all evaluated intervertebral levels.

Figure 5 indicates the IROM and ΔST scores of the participants. The IROM score was 0 in 75% of participants. ΔST score was ≥1 in approximately 70% of cases.

Table 3 shows the reliability of radiological parameter measurements. The mean intra-class correlation coefficient of intra- and inter-rater reliability for intervertebral disk angle was 0.73 and 0.66, and that for sagittal translation was 0.75 and 0.51, respectively. The inter-rater reliability of sagittal translation at L1–2 in extension and at L5–S in flexion was slight and fair, respectively, based on the intra-class correlation coefficients [25].

Complete data of participants is shown in Appendix A.

## 4. Discussion

Radiological lumbar instability has been thought to be related to LBP [7]. Various diagnostic criteria have been proposed for lumbar instability [18,19,26,27,28,29]; however, the relationship between lumbar instability and LBP remains uninvestigated [7,8,14]. Some studies have measured radiological parameters of the lumbar spine in healthy volunteers [13,14,15,16,17], but they did not evaluate ADL impairment using patient-oriented tools for assessing LBP-related ADL disabilities. The literature revealed that benchmark measurements for lumbar instability are used dating back more than three decades without validation with many healthy individuals [18,19,26,27,28,29].

To the best of our knowledge, this is the first study to collect and analyze systematic measurement data from dynamic lumbar radiographs of many healthy individuals. ODI and RDQ are widely used patient-oriented tools for assessing LBP-related ADL disabilities. In general, ODI of ≥20% and RDQ of ≥14 points are considered to indicate LBP-related impairment [22,23]. Participants exceeding the aforementioned ODI and RDQ thresholds were excluded from this study to assess lumbar dynamics in individuals without LBP-related ADL impairments. Table 4 lists IROM and ΔST measurements reported in other studies. IROM in our cohort was smaller, whereas ΔST was comparable to the previously reported values. The observed discrepancy in IROM compared with some earlier studies can be explained by the fact that participants were younger or were seated during dynamic radiography in the latter. Our study participants were older and were standing during the radiological examination. Conversely, ΔST is consistent between sitting and standing positions [30], which supports the equivalence of ΔST between the present and previous studies. Dvorak et al. [17] revealed that passive lumbar flexion rather than active flexion may have caused a larger IROM than that observed in our cohort.

Nachemson’s criteria for lumbar instability, i.e., IROM of ≥10° and ΔST of ≥3 mm at L1–2 to L4–5, and IROM of ≥20° and ΔST of ≥4 mm at L5–S, have been widely adopted [18]. White et al. proposed a threshold IROM of 15° at L1–2 to L3–4, 20° at L4–5, and 25° at L5–S, with a ΔST of 4.5 mm at each level [19]. However, these criteria were not supported by data from a sufficient number of patients or healthy volunteers. The present study established standard values of IROM and ΔST that were measured and calculated from many healthy volunteers. The threshold values for IROM at L1–2 to L4–5 were 10.3–12.4° in our cohort, which were higher than Nachemson’s cut-off value of 10° and did not exceed White’s cut-off value of 15° or 20°. The threshold values for ΔST at L1–2 to L4–5 were 5.4–5.5 mm, which exceeded Nachemson’s cut-off value of 3 mm and White’s cut-off value of 4.5 mm. Conversely, the threshold values for IROM and ΔST at L5–S were 13.9° and 3.3 mm, respectively, which do not exceed Nachemson’s (20° and 4 mm) and White’s cut-off values (25° and 4.5 mm). Therefore, the instability criteria advocated by Nachemson could not be necessarily suitable except for the L5–S level. As a matter of course, the standard upper limits of IROM and ΔST need to be verified in subjects with lumbar spinal diseases before they can be used to define lumbar instability in pathological situations because we only included asymptomatic participants without lumbar movement restrictions.

Moreover, we evaluated the number of intervertebral segments exceeding the reference values of IROM and ΔST using IROM and ΔST scores according to Nachemson’s criteria. These scores indicate that 25% of healthy participants meet the IROM criteria at any intervertebral level, and 7.6% had excessive angular motion at two or more levels. Conversely, 72% of participants met the criteria for abnormal ΔST at any level, and up to 41% deviated from Nachemson’s criteria at two or more levels [18].

Iguchi et al. revealed that ΔST has a greater influence on lumbar symptoms than IROM; however, asymptomatic individuals in our cohort demonstrated ΔST instability more frequently than IROM. This discrepancy is due to patient selection. Various symptoms were contained rather than LBP because the patients presented in the former report had lumbar spinal canal stenosis [24]. Hayes et al. concluded that >3 mm of ΔST is questionable as an indicator for fusion surgery because of lacking evidence. They also mentioned that the usage of dynamic radiographs of the lumbar spine for lumbar instability assessment is problematic. This opinion is in concordance with our result [15].

The relationship between LBP and radiological lumbar instability remains controversial. Our study could not evaluate the relationship between LBP and radiological lumbar instability as our cohort did not include participants with intermediate to severe LBP; therefore, additional studies of patients with severe LBP are necessary to clarify this association.

The inter-rater reliability of the intervertebral disk angle measured in our study was higher than that of sagittal translation. The lower reliability of sagittal translation measurements can be associated with the difficulty of controlling the obliquity of the X-ray beam and the influence of soft tissues surrounding the lumbar spine [31]. The posterior aspect of the vertebral body appears as two lines when the X-ray incidence angle is not parallel to the vertebral body, causing inaccurate sagittal translation measurement [32].

This study has several limitations. Most of the volunteers were schoolteachers; therefore, the cohort could have been biased toward non-manual laborers in terms of sample selection, which can affect the prevalence of lumbar degeneration and/or related symptoms. Also, the relationship between excessive mobility of the lumbar spine and LBP could not be evaluated because patients with severe LBP were excluded. Additionally, the degree of active flexion and extension depended on each participant as opposed to maximum passive motion. However, this study provides the first set of precise measurements for lumbar mobility in a large sample of participants who had less LBP-related disability according to ODI and RDQ. This is a cross-sectional study focusing on a single time point, so further cohort studies could confirm the concept in the future.

## 5. Conclusions

The normative data of lumbar mobility were obtained from a large number of participants who had less LBP-related ADL disability, according to ODI and RDQ. The widely accepted criteria for lumbar instability were not applicable except for the L5–S level. Further study of lumbar mobility, including patients with severe LBP, might prove that a higher threshold should be established or that hypermobility is not related to LBP.

## Figures and Tables

**Figure 1 jcm-13-06116-f001:**
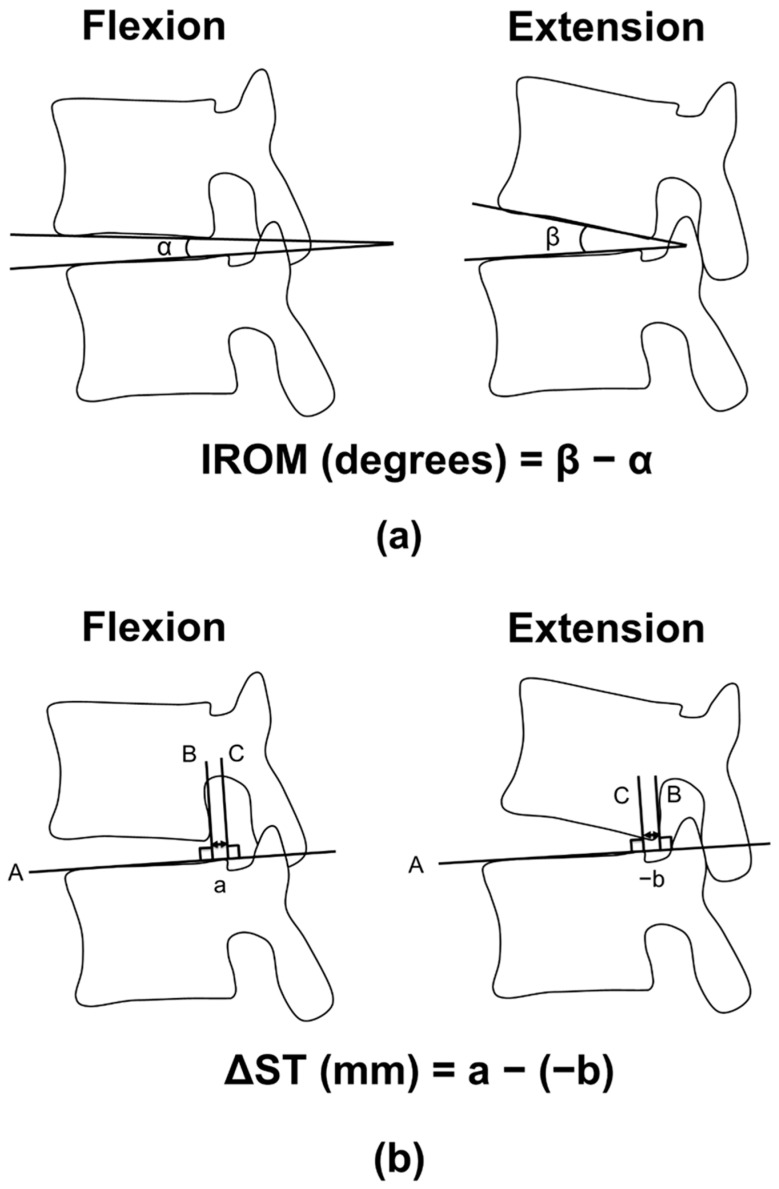
Measurement methods of the intervertebral range of motion (IROM) and the sagittal translation distance (ΔST): (**a**) For the measurement of IROM, two lines were drawn along the lower endplate of the upper vertebral body and the upper endplate of the lower vertebral body. Then, the angle made by these two lines (intervertebral disc angle) was measured in flexion and extension, respectively. The intervertebral disc angle was counted positive when the wedge made by two lines was opened ventrally. IROM was given by the angle of the intervertebral disc angle in extension subtracted by that in flexion. (**b**) Sagittal translation between adjacent vertebrae was defined as follows [24]. First, a line was drawn on the upper endplate of the lower vertebral body (Line A). Then, two lines perpendicular to this line touching the postero-inferior edge of the upper vertebra (Line B) and the postero-superior edge of the lower vertebra (Line C) were drawn, respectively. The sagittal translation was defined as the distance between those two lines. If the upper vertebra was located anterior to the lower vertebra, the sagittal translation value was positive. ΔST was given by the difference of the sagittal translation in flexion subtracted by that in extension.

**Figure 2 jcm-13-06116-f002:**
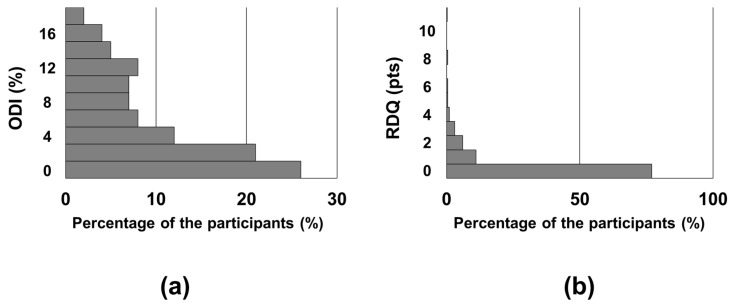
The histograms of the Oswestry Disability Index (ODI) and the Roland–Morris Disability Questionnaire (RDQ): (**a**) The ODI scores were less than 6% in about 60% of the participants. (**b**) The RDQ scores were 0 points in 77% of the participants.

**Figure 3 jcm-13-06116-f003:**
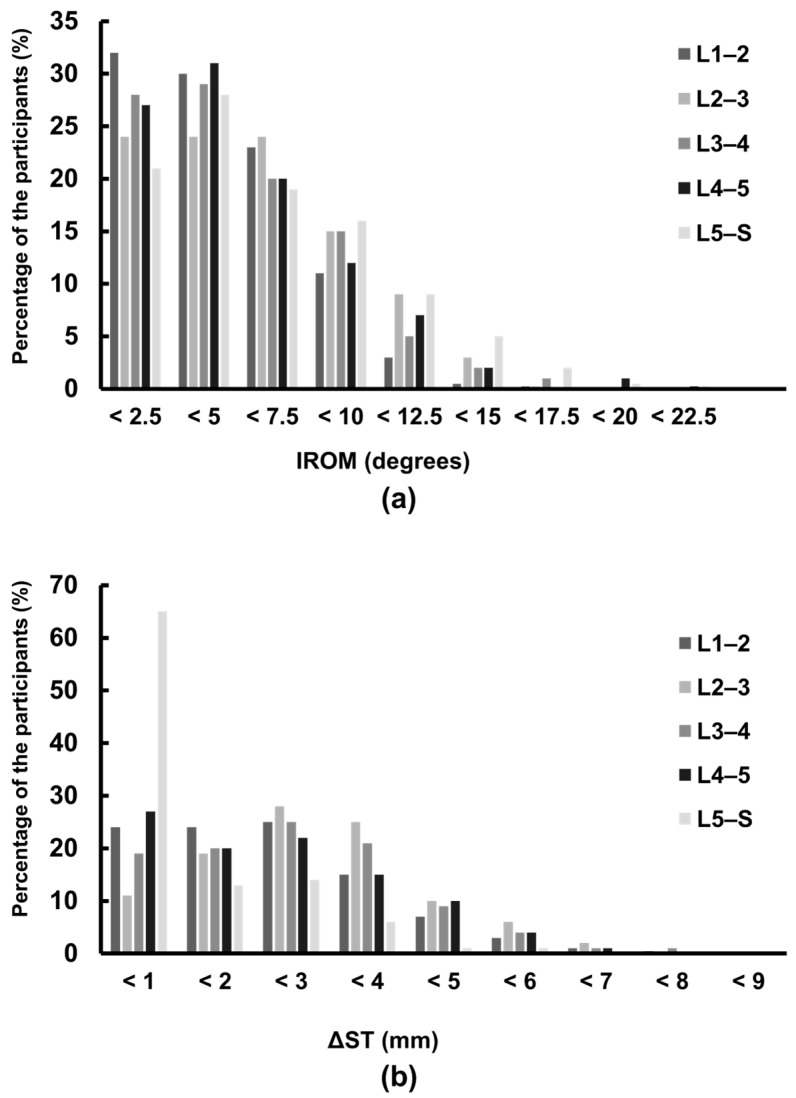
The histograms of the intervertebral range of motion (IROM) and the sagittal translation distance (ΔST) in each intervertebral level: (**a**) At the L1–2 to L4–5 level, 3.8–12.1% of the participants showed IROM more than 10 degrees. (**b**) At the L1–2 to L4–5 level, 28–43% of the participants showed ΔST more than 3 mm.

**Figure 4 jcm-13-06116-f004:**
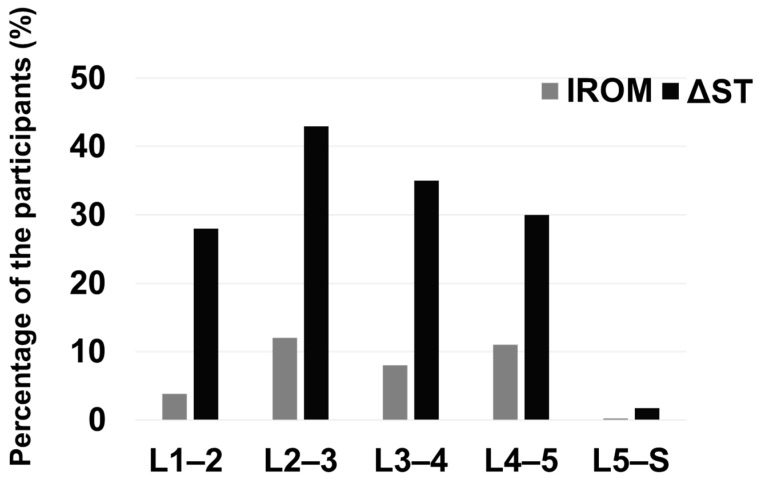
The percentage of the participants who exceeded the limit of lumbar instability at each intervertebral level: More than 40% of the participants exceeded the threshold of the sagittal translation distance (ΔST) at the L2–3 level. Only 0.2% and 1.7% of the participants showed lumbar instability at the L5–S level regarding the intervertebral range of motion (IROM) and ΔST, respectively.

**Figure 5 jcm-13-06116-f005:**
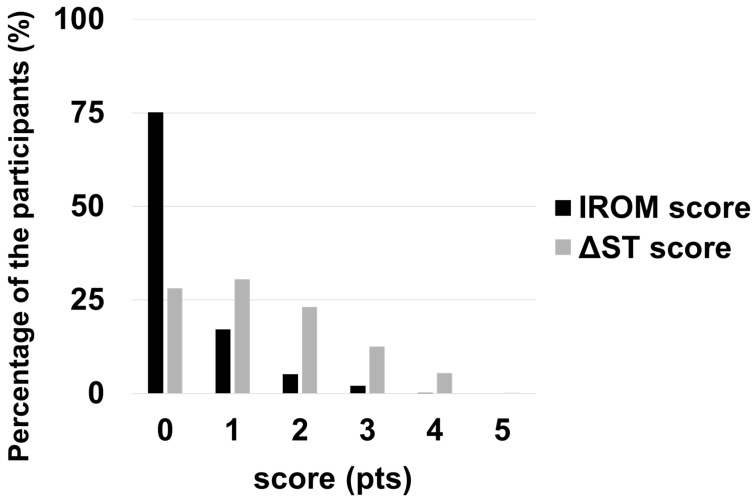
The intervertebral range of motion (IROM) score and the sagittal translation distance (ΔST) score of the participants in bar graphs: For quantitative evaluation of the lumbar instability, the number of intervertebral levels fulfilling the criteria of radiological lumbar instability of IROM and ΔST was defined as the IROM and the ΔST score, ranging from 0 to 5, respectively. The IROM and the ΔST scores were one or more in 25% and 72% of the participants, respectively.

**Table 1 jcm-13-06116-t001:** Demographics of the participants.

Number of Participants	420
Sex (M:F)	249:171
Age (years)	55 (24–83)
Height (cm)	167 (144–191)
BW (kg)	66 (39–142)
BMI	23.7 (16.4–41)

BW, body weight; BMI, body mass index. Data are shown in median (range).

**Table 2 jcm-13-06116-t002:** Mean, standard deviation, and standard upper limit of IROM and ΔST at each intervertebral level.

	IROM			ΔST		
	Mean	Standard Deviation	Standard Upper Limit [95% CI]	Mean	Standard Deviation	Standard Upper Limit [95% CI]
L1–2	4.3	3.0	10.3 [9.8–10.8]	2.2	1.6	5.4 [5.1–5.7]
L2–3	5.4	3.5	12.4 [11.8–13.0]	2.7	1.4	5.5 [5.3–5.7]
L3–4	4.9	3.4	11.7 [11.1–12.3]	2.4	1.5	5.4 [5.1–5.7]
L4–5	5.0	3.6	12.2 [11.6–12.8]	2.2	1.6	5.4 [5.1–5.7]
L5–S	5.9	4.0	13.9 [13.2–14.6]	0.9	1.2	3.3 [3.1–3.5]

IROM, intervertebral range of motion; ΔST, distance of sagittal translation. The standard upper limit is the upper limit of 95% reference interval. The 95% CI is the 95% confidence interval of the standard upper limit.

**Table 3 jcm-13-06116-t003:** Intra- and inter-rater measurement reliability of the radiological parameters.

	Intra-Rater Reliability	Inter-Rater Reliability
Intervertebral disc angle		
L1–2 flexion	0.708	0.684
extension	0.648	0.512
L2–3 flexion	0.789	0.789
extension	0.672	0.602
L3–4 flexion	0.857	0.810
extension	0.665	0.486
L4–5 flexion	0.819	0.798
extension	0.768	0.731
L5–S flexion	0.630	0.518
extension	0.785	0.694
Mean	0.734	0.662
Sagittal translation		
L1–2 flexion	0.604	0.443
extension	0.756	0.167
L2–3 flexion	0.775	0.561
extension	0.786	0.588
L3–4 flexion	0.858	0.515
extension	0.840	0.720
L4–5 flexion	0.854	0.663
extension	0.892	0.706
L5–S flexion	0.574	0.261
extension	0.564	0.502
Mean	0.750	0.513

**Table 4 jcm-13-06116-t004:** Lumbosacral spine motion in normal individuals.

Authors	Number of Subjects	Sex	Age	Positioning during Dynamic View	IROM (Degrees: Mean)					ΔST (mm: Mean)				
					L1–2	L2–3	L3–4	L4–5	L5–S	L1–2	L2–3	L3–4	L4–5	L5–S
Clayson et al. (1962) [13]	26	F	College Students	Sitting, active flexion/ Standing, extension	12.6	15.8	15.9	17.7	18.7	NA	NA	NA	NA	NA
Pearcy et al. (1984) [14]	11	M	25–36	Standing, active flexion–extension	13	14	13	16	14	4	3	3	3	2
Hayes et al. (1989) [15]	59	M	19–59	Sitting, active flexion–extension	7	9	10	13	14	1.9	2.4	2.5	3.0	1.3
Boden et al. (1990) [16]	40	M	19–43	Standing, active flexion–extension	8.2	7.7	7.7	9.4	9.4	1.4	1.3	1.2	1.2	1.0
Dvorak et al. (1991) [17]	41	M/F	22–50	Standing, passive flexion–extension	11.9	14.5	15.3	18.2	17	2.6	3.0	3.1	2.6	0.9
This Study	420	M/F	24–83	Standing, active flexion–extension	4.3	5.4	4.9	5.0	5.9	2.2	2.7	2.4	2.2	0.9

IROM, intervertebral range of motion; ΔST, sagittal translation distance; NA, not available.

## Data Availability

The datasets generated during and/or analyzed during the current study are available from the corresponding author upon reasonable request.

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
