# Peer review of "Validating the Definition of Lumbar Instability—A Cross-Sectional Study with 420 Healthy Volunteers"

_jcm, 2024, doi:10.3390/jcm13206116_

Round 1
Reviewer 1 Report
Comments and Suggestions for Authors
This manuscript makes an important contribution to understanding lumbar instability, particularly in the context of the need to standardize diagnostic criteria. The methodology of the manuscript is solid, with clear definitions and strict inclusion criteria for participants.
Although the study significantly contributes to understanding radiological lumbar instability, there are several potential limitations that could affect its conclusions:
· Sample Non-Selectivity: The majority of the participants in the study (75%) were school teachers. Such a demographic composition may limit the generalizability of the findings to the broader population since professions with specific workloads and lifestyles may differ in terms of the prevalence of lumbar issues.
· Focus on the Healthy Population: While establishing norms for a healthy population is important, the study did not sufficiently include participants with pronounced lower back pain (LBP). This makes it difficult to assess the potential threshold of instability that would be clinically significant for patients with LBP.
· Limited Validation of Criteria: Although Nachemson's criteria were used, the study did not provide a detailed analysis of whether these criteria are truly the best indicators of lumbar instability. The research focused on the thresholds of IROM and ΔST but did not explore how these criteria correlate with actual symptoms and functional outcomes.
· Limited Technical Application of Radiography: The study used standard dynamic radiography but did not consider advanced imaging techniques (e.g., MRI), which could offer a more precise analysis of the lumbar segment and its stability.
· Short-Term Study: The study was conducted over a short period (from January to June 2018), which may limit the ability to track long-term changes in lumbar stability in healthy participants.
· Outdated Sources: Most of the sources cited in this manuscript are outdated, with 25 out of 26 references being over five years old.
Reviewer 2 Report
Comments and Suggestions for Authors
I found the information presented very interesting and the content of the manuscript useful. Please find the attached PDF with my comments inserted. These are meant to assist in the refinement of the manuscript. Overall, I thought the document was positive and has a firm foundation from which to build.

Round 2
Reviewer 1 Report
Comments and Suggestions for Authors
I suggest that the article be accepted for publication.